# Designing profitable and climate-smart farms using virtual reality

Remy Lasseur[1]*, Seth Laurenson[1], Mohsin Ali[2], Ian Loh[2], Mike Mackay[1]

1 AgResearch, Lincoln Research Centre, Private Bag 4749, Christchurch, New Zealand, 2 Mohsin Media Designer, Wellington, New Zealand

* remy.lasseur@agresearch.co.nz

## Abstract

Many pastoral farmers are searching for ways to lower the carbon emission footprint that is generated by livestock. Planting trees on the farm is currently a popular option for farmers to offset their emissions yet requires knowledge of suitable tree species and locations to plant them. This paper describes a decision-support tool aimed at helping farmers to create and visualise different planting designs while balancing the objectives of sequestering carbon and maintaining farm profitability. We take an innovative approach by combining virtual reality technology with biophysical models to create an environment where the user can actively create virtual future farm scenarios. Through the creation process, the user can simultaneously balance multiple objectives including farm aesthetics, economic returns, business and environmental ambitions, and carbon emissions (net) balance. For this proof-of-concept study, we incorporate virtual reality technology in *Unreal Engine*, environmental and financial data, and high-resolution spatial layers from an operational 400-hectare livestock farm in New Zealand.

## Introduction

Global agriculture is facing a significant challenge as it transitions towards greater sustainability [1]. There is a strong expectation that digital technology will have an important role in supporting farmer decision-making at both an operational and strategic level as they transition to more sustainable food production systems [2]. In New Zealand, a growing array of digital tools are available to support farmers in making day-to-day business decisions. However, generally these tools are limited in functionality that enables users to interact with the underlying scientific models and databases [3]. In this paper we describe the development of an innovative virtual reality (VR) application that combines agricultural economics, environmental metrics, and VR. The tool is targeted at livestock farmers who are considering afforesting all or some of their farmland. Our opening assumption was that the integration of VR would add a new and valuable dimension to existing farmer decision support tools because landowners are able to see, explore and compare (visually and quantitatively) future planting strategies. This assumption has been informed by some recent examples in the literature [4–6].

Our project is relevant in New Zealand because farmers are currently adapting to a several recent regulatory changes that are providing the impetus to consider new land uses. For

**Data Availability Statement:** All relevant data are within the manuscript and its Supporting Information files.

**Funding:** This project was funded by the New Zealand Ministry for Business, Innovation and

Employment through AgResearch's research programme 'New Zealand Bioeconomy in the Digital Age'. The funders had no role in study design, data collection and analysis, decision to publish, or preparation of the manuscript.

**Competing interests:** The authors have declared that no competing interests exist.

example, the New Zealand Government's *Climate Change Response Amendment Act 2019* will require farmers to reduce all greenhouse gasses (except biogenic methane) to zero, and biogenic methane emissions by more than 24% (compared to 2017 levels) by 2050. Achieving the target is expected to have a significant impact on livestock farms because approximately half of national carbon emissions originate from animal farming [7]. One option for New Zealand farmers is to plant trees across parts of their farm with the intention of offsetting their carbon emissions and potentially earn credits via New Zealand's 2008 *Emissions Trading Scheme* (ETS) [8, 9].

Large scale conversion of farmland to forestry following escalating global carbon credit prices has generated anxiety within rural communities [10]. Some commentators have argued that the replacement of livestock farms with forest will lower community resilience and vitality [11]. To date large-scale land conversion to forestry has been primarily confined to low-producing farms that were previously generating low financial returns [10], new regulatory settings and a high carbon price are expected to encourage the conversion of highly productive land into trees [9].

This paper describes the development of a VR application that is designed to help livestock farmers create an *integrated* forestry-livestock system. We address the current challenge livestock farmers face to identify the most suitable locations on the property where trees can be planted so that they can meet their emissions objective (i.e., a carbon neutral enterprise) without compromising the profitability of their businesses, or affect amenity values of the property (such as aesthetic amenities, recreational access or sacred indigenous sites). In practice, the VR application enables farmers to 'stand in their farm' (a virtual environment) and virtually plant different tree species in different locations. Different tree species have different carbon sequestration rates, aesthetic, and establishment costs. Therefore, the impacts of tree planting depend on the selected tree species and the planting location. Farmers can subsequently, visualise the resulting landscape, and assess what this means for farm profitability and their emissions targets. It also allows the comparison of different tree planting strategies. Our application has been built to a to prototype stage.

The project brought together a multidisciplinary team with capabilities in VR, remote sensing, farm economics, carbon accounting and product design. The team produced a high-resolution landscape visualisation that enables the user to identify planting areas based on the spatial variability of pasture production and in doing so, quantify the carbon sequestration and profitability outcomes from different planting scenarios. The VR environment also enables users to visualise the aesthetic changes made to the farm under each scenario. The prototype described in this paper has been developed in Unreal Engine® and is set on an existing 400 ha New Zealand beef and sheep farm in Canterbury, New Zealand.

Several digital decision support tools are available in New Zealand that can be used to assist farmers to design carbon neutral farm enterprises [e.g., 12, 13]. We also acknowledge the presence of technical guidance documents that are freely available in New Zealand to help farmers understand the implications of converting land to forestry, such as look-up tables and technical guides. However, our new application complements the existing information by adding VR technology, a novel communication style seldomly used to convey scientific data [14]. We take this approach because the ability of humans to understand and assimilate information from pictures is greatly enhanced compared to conveyance through traditional scientific representations such as graphics and numbers [3, 15–17]. Therefore, we believe our approach will support more effective conversion of information into user knowledge and facilitate the implementation of new practices. So far, the use of visualisation in landscape planning has largely been limited to a supporting role for participatory processes or to help policy making [18, 19].

This paper focuses on the technical development of the application, beginning with a description of the methods we used to model farm profits and carbon emissions. The next section describes the study site and the technical modelling that we used to predict carbon emissions and profitability. The next section also describes the method we used to develop the VR environment and to visualise tree plantings and the impact of the planting regime on farm performance (carbon, profitability). We then present the outcomes delivered by the VR application and provide some reflections about how the tool might support more interactivity between users and scientific data used for farm design and to achieve carbon targets.

## Study site and methods

### Study site

The VR tool was developed using a 400-ha sheep and beef farm located on Banks Peninsula, Canterbury, New Zealand (Fig 1). In choosing a pilot farm we looked for an operating livestock farm with a diverse terrain that had the potential for tree planting. We also searched for a location that had high amenity value, in this case unabated views of the Pacific Ocean and access to recreational sites. At the time this study was initiated, the farm owners were embarking on a re-forestation programme that aimed to expand the area in native trees at the cost of the current pastoral land.

The farm's terrain was characterised as mixed rolling downlands and steep gullies covered predominantly in pasture. There were small areas of exotic pine (*Pinus radiata*) and regenerating native forest (including *Prumnopitys taxifolia* and *Podocarpus totara*). The farm is bordered on three sides by livestock farms, while the southern boader met the Pacific Ocean. The farm

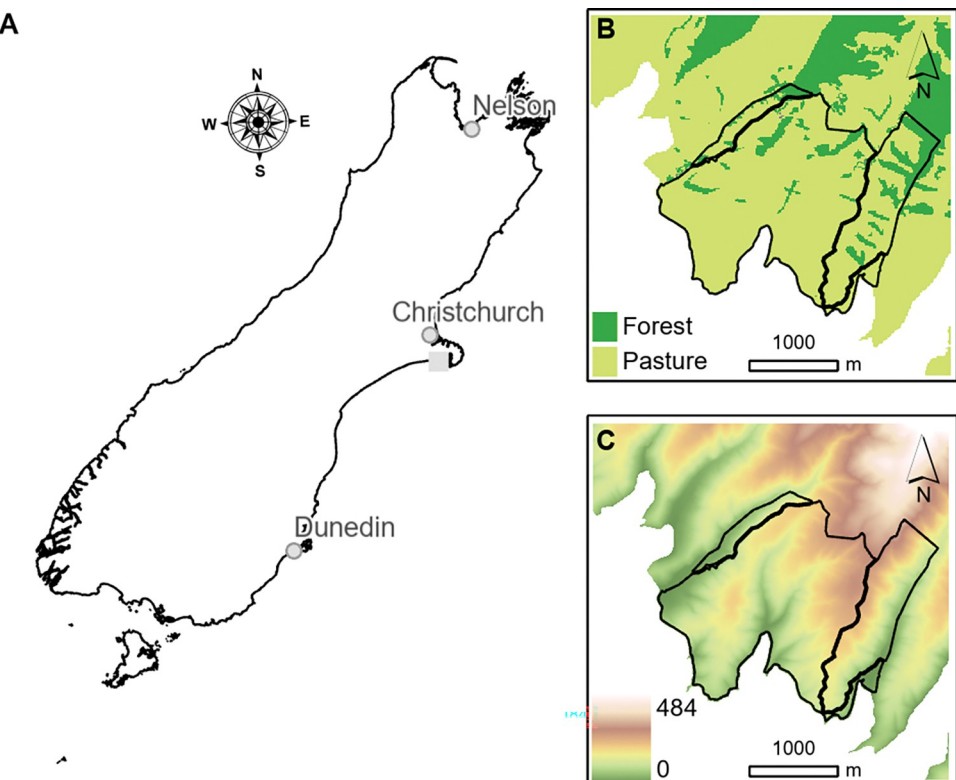

**Fig 1. Location of the study site.** (A) Location of the farm on the South Island of New Zealand, (B) Farm boundaries and the land cover map. Light green represents beef and sheep pasture, dark green represents native forest and brown represents Pine trees. (C) Elevation map from the sea level to 690m.

currently sustains between two and three stock units per hectare, which is below the 6.3 stock unit per hectare in average for New Zealand sheep and beef farms [20]. As such, we chose to adjust the stocking rate to match the industry average, so that the VR application we were developing was more broadly applicable to the national context and challenges of livestock farms.

## Data sources

**Geospatial data.** Geospatial data have been used to map the spatial variation of the pasture production and to generate the realistic virtual landscape. Normalised Difference Vegetation Index (NDVI) data from Sentinel-2 images have been used to classify the relative pasture production. To do so, two satellite tiles from the 2019 lambing season (October and November) have been accessed through the EOS Data Analytics website [21]. To create the geospatial 3D terrain within the virtual environment, a one-meter spatial resolution Digital Elevation Model (DEM) has been downloaded from the Land Information New Zealand website [22] and comes from Lidar data captured between 2018 and 2019. A 3-band (RGB) orthophotography with 300 mm spatial granularity taken in October and November 2020 was obtained from Land Information New Zealand website [23] and was used to generate a base map that was displayed in the virtual environment (described below).

**Carbon sequestration data.** Carbon sequestration values for the different tree species represent the values that are outlined in *New Zealand's Climate Change (Forestry Sector) Regulation 2008* (SR 2008/355), available from the New Zealand Ministry for Primary Industries [24]. These sequestration values are available for three tree species including native trees (a broad category encompassing a range of New Zealand native species), Pine (*Pinus radiata*) and Eucalytpus (*Eucalyptus nitens*). The amount of carbon that is sequestered by trees varies depending on age. However, in this study, we have considered the yearly average sequestration over a 30-year period. The establishment costs for the different types of trees were based on expert knowledge and, for each individual species, a uniform rate was applied (i.e., the cost of planting on sloping terrain was assumed to be the same as planting on flat terrain).

**Carbon emission and profitability data.** Carbon emission and finance data have been extracted from existing industry literature. Carbon emissions and financial profitability vary spatially across the farm according to stocking rate (e.g., number of animals per hectare). Therefore, carbon emissions and profitability values have been expressed per Stock Units (SU), respectively kgCO2eq/SU/year and NZD/SU/year.

Gross carbon emissions from livestock farming primarily arise from methane generated by enteric fermentation in the rumen [25]. Between the years 2018 and 2020, the national SU average for the sheep and beef industry ranged from 41.9 million to 42.5 million with an average of 42.2 [26]. Livestock emissions estimates for the same years have ranged from 13,551 kt to 16,537 kt [26, 27]. These estimates suggest a gross carbon emission ranging between 327 and 395 kgCO2eq/SU/year for New Zealand sheep and beef farms. An emission of 370 kgCO2eq/SU/year has been assumed.

In this study, profitability is defined as the Earnings Before Interest, Tax and Rent (EBITRm). Profitability varies from year to year and depends, amongst other things, on market and climatic conditions. Data from the literature suggests that the average national profitability for beef and sheep farms in New Zealand ranges between $42 and $62 per SU [20]. In the current study, NZD $56/SU has been assumed.

## Methods

The main objective of the VR application was to enable farmers to explore different tree planting strategies on their farms. Within the VR environment, the user can virtually 'plant' one of

three forest types that are commonly grown on New Zealand farms. The following sections describe the methods used to estimate the impacts of tree planting on the farm profitability and carbon emissions.

## Virtual reality application

The VR application was developed using *Unreal Engine* (UE, version 4.26.2) and designed to run on an *Oculus Quest 2* headset. *UE* provides a blueprint visual scripting function that enables the development of VR applications with limited coding which in turn accelerates the development phase and reduces costs [28]. The virtual landscape within UE was made using a grid of 4033*4033 vertices, each covering one square meter, which is the recommended landscape size to optimise headset performance. Therefore, the area considered in the application is covering 1,600 ha centred on the pilot farm. So, the user can navigate through the farm of interest and its surrounding. To improve the performance, the application was developed to be run with the VR headset connected to a high-performance computer.

A high-resolution height map was generated in ArcGIS Pro (version 2.5.0) using a one-meter resolution DEM, which was converted from Geotiff to PNG format and imported into UE. The 300 mm aerial imagery was resampled to 500 mm spatial resolution and imported as a PNG file into UE. This 500mm aerial photo was 'draped' on top of the 3D terrain to create landscape texture. The map of pasture production, based on the NDVI classification, was also imported (as a PNG file) into UE as an overlay to the landscape texture and displayed with high transparency. This enables the user to visualise simultaneously the landscape texture and the pasture production categories. At the end of a session, a map representing the spatial footprint of the planting done by the user can be exported and stored as a *jpeg* image. Alongside the planting map, screenshots of the virtual environment can also be saved and exported as jpeg to support the user in communicating new planting designs.

## Carbon emissions and financial modelling

Within the VR twin of the farm and using the VR headset and hand controls, the user selects 30m x 30m grid cells where trees can be located. When the user validates a planting site, the whole farm profitability is estimated by Eq (1).

$$Farm\ profit = Ps - (Pp + Ct) \qquad \text{Eq (1)}$$

Where; farm profit (NZD/year); $P_s$ is farm profit before planting (NZD/year); $P_p$ is the profit that was generated by the livestock operations on the selected grid cell prior forestry conversion (NZD/year) and $C_t$ is the establishment cost of the planted species (NZD).

The overall farm carbon emissions are estimated in the application by Eq (2).

$$Farm\ emissions = Es - (Ep + St) \qquad \text{Eq (2)}$$

Where farm emissions (kgCO$_2$eq/year); $E_s$ is the farm emissions before planting (kgCO$_2$eq/year); $E_p$ is the emissions generated by the selected grid cell prior to being converted to forest (kgCO$_2$eq/year) and $S_t$ is the sequestration generated by the planted tree species (kgCO$_2$eq/year).

## Pasture production and stock units

The profit and the carbon emissions arising from a given area of pasture (respectively Pp and Ep in Eqs 1 and 2) depends on the stocking rate which varies across grid cells according to the pasture production (i.e. the feed resource). Based on the distribution of the NDVI across the

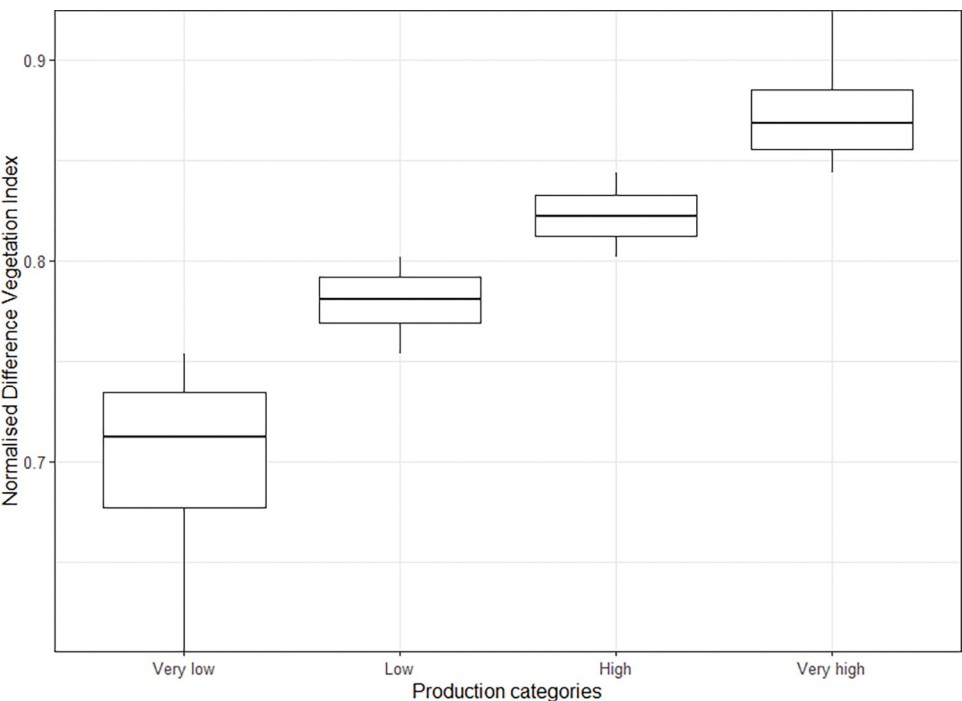

**Fig 2. Distribution of the Normailsed Difference Vegetation Index (NDVI) values for the categories of pasture production.**

farm (quartile groupings), all the grid cells covered by pasture have been assigned to one of four categories ranging from very low to very high productive land (Figs 2 and 3). For the very low and very high producing categories, we have used a stocking rate of 1.8 and 10.4 SU/ha, respectively, which is within the range observed across hill country sheep and beef farms in NZ [29]. We have used intermediate stocking rates for the two remaining classes

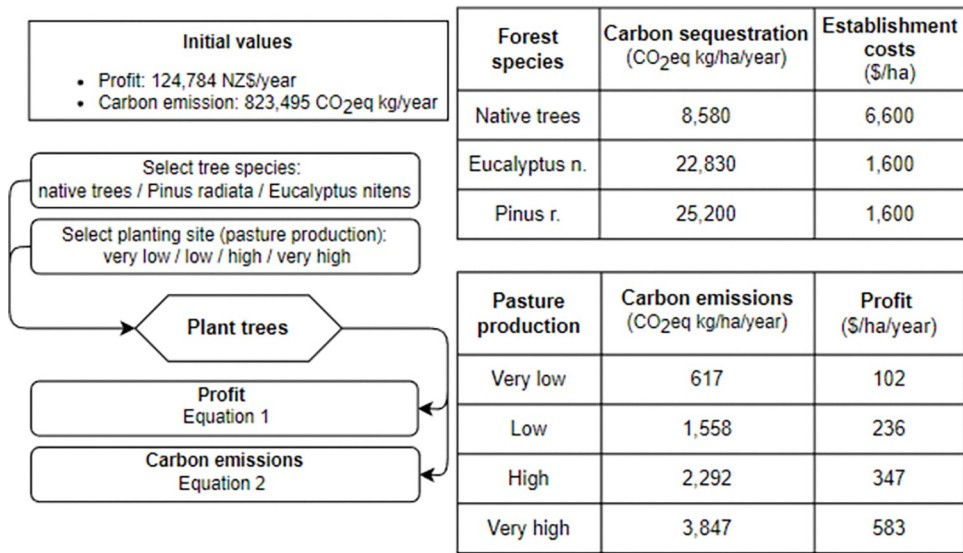

**Fig 3. Application logic and values used for the different forest types and pasture relative productivity classes.**

(approximately 4.6 and 6.5 SU/ha for the low productive and high productive categories, respectively). Carbon emissions and profit have been distributed across the four pasture categories based on stocking rate assumptions and the profit and carbon emission rate described previously (Fig 3).

The initial greenhouse gas emissions ($kgCO_2eq$ /ha) and profitability (NZD/ha) balance were established by considering the existing land use of the farm. Regarding profitability, the initial farm profit before interest, tax and rent has been set to $124,784 per year (Fig 3), indicating an average profitability of $317/ha pasture. This value is representative of NZ beef and sheep farm profitability that has ranged from $266 to $401 per hectare for the season 2019/2020 and 2020/2021 respectively [20]. The initial carbon balance has been set up to 823,495 $kgCO_2eq$ per year.

## Results and discussion

Using the pilot farm as a test case, we now report the findings of our experimentation with the tool and its ability to compare different tree planting strategies based on profit, GHG and a visual appreciation/assessment of the tree (fully grown). One of the most challenging aspects of the work, but one of great importance and highlighted next, was our need to understand the spatial variability of pasture production across a property. For this project, it was important because we assumed that farmers will be looking to plant trees on areas of the farm where pasture productivity is low and consequently profit is also. A second challenge for us was to create a virtual environment that accurately represented the farm. Our assumption here is that farmers will expect to be able to move across their whole property and that the planting strategies that they design in the tool provide a realistic impression of the changes 'in the landscape' alongside environmental and financial targets, giving them added confidence in their final decisions. We note here that farmers have a great deal of (tacit) knowledge about their properties that is relevant for strategic planning, so the application was developed with this in mind, providing a bridge between modelling science and farmer experience. This latter point features in our discussion.

### Data integration

The farm terrain is a mix of steep hill country and flat areas that extend across the tops of the hills and within large gullies (Fig 4A). Steep hill country comprises a large portion of the farm

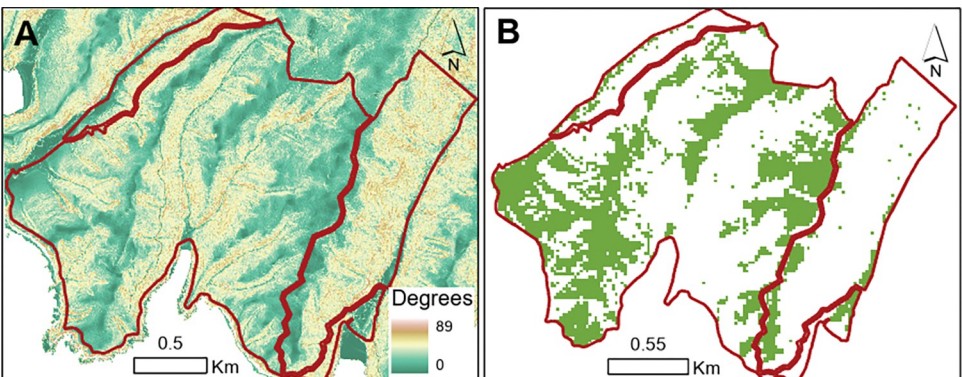

**Fig 4. Slope and very high pasture production areas.** (A) Map of the farm border (red line) with the slope. Yellow and bright colour indicates steep slopes while light green colour indicates flat areas. (B) Map of the farm border (red line), with the most productive areas highlighted in green. Relative grass productivity has been estimated based on two NDVI images from Sentinel-2 taken during the 2019 lambing season.

with half of the total farm area being on slopes that are 27 degrees or more, while less than 10% of the farm is located on slopes between 0 and 10 degrees.

The pasture-productivity map (Fig 4B) derived from NDVI data locates the highest productive pasture on the flat areas along the hill tops and the flat areas in between the steep gullies that extend along a North to South plane. Other areas of the farm have low and very low pasture productivity, most likely due to combined factors such as the steep slope, shallow soils and low soil fertility which is typical of landscapes in this region.

As expected from the maps above, there is a negative trend between the NDVI values and slope i.e., pasture productivity tends to be greatest in flat areas (Fig 5). Considering the carbon and profit values used in this study (which reflect the national average) our model highlighted that 8% of the farm would need to be converted to forestry if the user only considers planting pine trees on very low productivity classes. This number increased to 22% of the farm if the user only consider planting native trees (Fig 6A).

Future iterations of this tool could reproduce the approach of previous studies that have used remote sensing data to generate robust estimates of pasture productivity over large areas [30–32]. This more in-depth analysis might also support a refinement of the stocking rates [33], farm financial returns and GHG emissions that would increase the level of detail considered into farm planning strategies. This approach was not used here, as the focus was on developing a prototype.

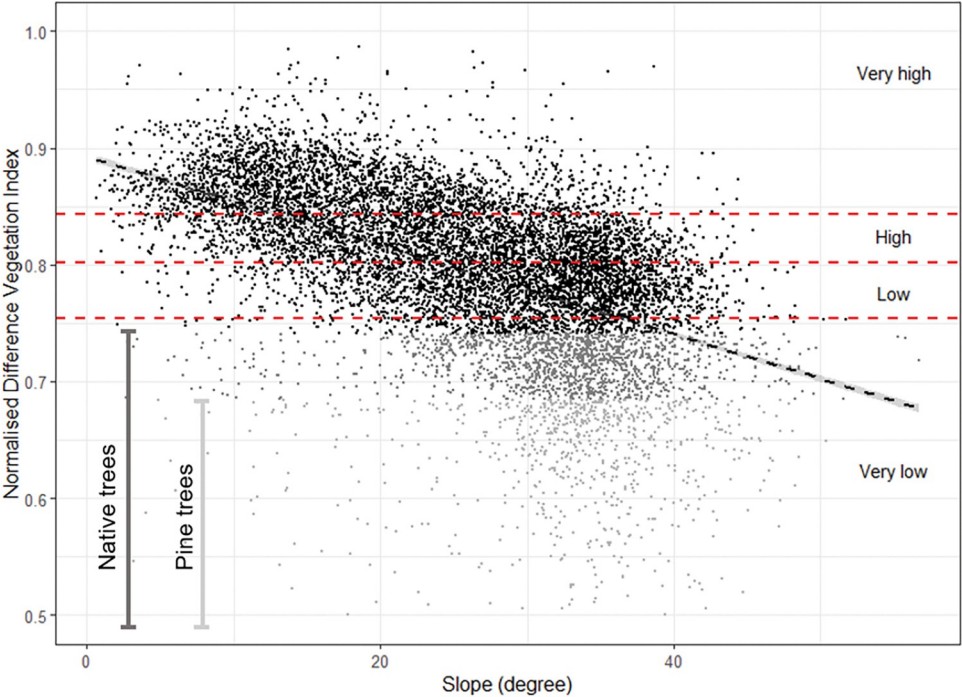

**Fig 5. NDVI values for sentinel-2 pixels (20m) according to the slope in degree.** NDVI values have been used as an estimate of relative pasture productivity (most producing pixels versus least producing pixels). The horizontal red dashed lines represent the four productivity quantiles used to classify the pasture productivity from very high productivity to very low productivity. As slope increases, pixel productivity tends to decrease. The black dashed line represents the linear equation describing this negative correlation between NDVI and slope. The grey shade colours (from light grey to black) used to represent the individual pixels within the scatter plot represent the pixels that would need to be converted from pasture to forest for the farm to be carbon neutral when converting the least productive pixels first. Light grey pixels (n = 798) up to NDVI values equal to 0.69 are the pixels that would need to be converted if only planting pine forest. Light grey pixels plus grey pixels (n = 2257) up to NDVI values equal 0.73 are the pixels that would need to be converted if only planting native forest.

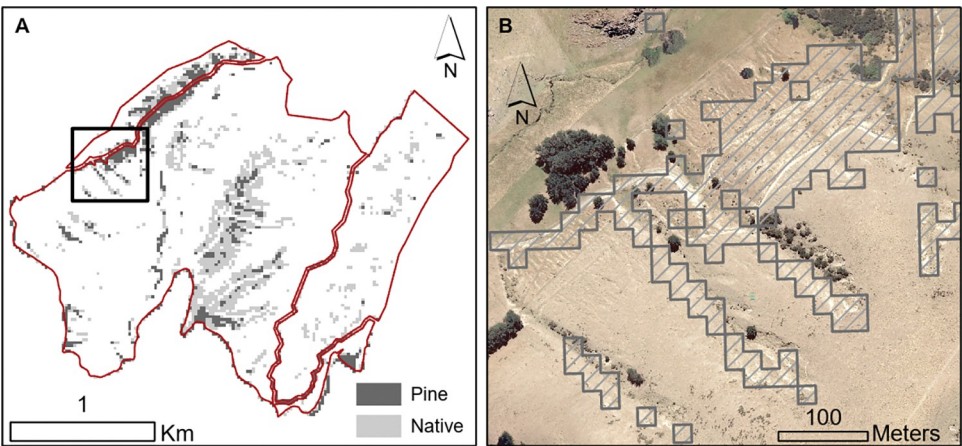

**Fig 6. Planting scenarios.** (A) Map of the farm border (red line). Map of the farm border (red line). Grey shaded areas represent the footprint (area) that would need to be planted for the farm to be carbon neutral with the assumption that the area producing the least pasture are converted first. When considering pine trees, only the dark grey area would need to be planted for the farm to be carbon neutral. When considering native trees, because their sequestration rate is smaller than pine trees, all the shaded areas (dark and light grey) would need to be planted for the farm to be carbon neutral. As expected from the results presented in Fig 5 the highlighted locations are mostly clustered around the steepest slopes of the farm. The black box at the northwest (top left) of the map highlights the spatial extent of the aerial imagery presented on (B). (B) High-resolution (300mm) aerial imagery of the location highlighted in (A). From a visual analysis, the bright yellow areas are bare soil on steep slopes suggesting very active land erosion that has also been confirmed during a visual assessment in the field. These areas overlap the planting zones suggested by the map on (A) and also represented by the hatched symbology on (B). Across the farm, converting the least productive areas from pasture to forest might help mitigating soil erosion in addition to generating carbon sequestration.

We have developed a linear logic model that combines carbon datasets with pasture production data so that the user can assess the interaction between GHG emissions and profitability under various scenarios. With future developments, a variety of similar science-based data (e.g., spatial layers, lookup tables, linear equations) can be incorporated into the VR application. For instance, tree planting strategies are expected to affect a range of other farm outcomes that are in addition to the two considered currently (i.e., carbon sequestration and farm profitability). Visual assessment of aerial imagery of the farm suggests there is significant overlap between areas that have the lowest estimated production (Fig 6A) and areas where there is evidence of soil erosion (Fig 6B).While planting trees in these locations will increase carbon sequestration with the least impact on profitability, mitigating hillslope erosion is also likely [34]. Furthermore, the ability to mitigate the impact of climate change on local surface temperature might also be considered in future steps [9, 35–40].

The planting design that maximises both carbon sequestration and profitability (i.e. Fig 6A), suggest a combination of small exotic forest patches distributed across the farm. However, under the current New Zealand ETS system, planted areas that are less than 1 ha are not eligible for carbon credits [41]. In this context, we believe that, in order to incentivise the simultaneous delivery of multiple ecosystem services by rural landscape and avoid the large-scale conversion of productive pasture to exotic forest, areas that are less than 1 ha should be made eligible for carbon credits.

The prototype application described here simulates the immediate consequences of the planting activities on farm cash flow, including loss in profit compared with the original land use and establishment costs. The sale of carbon credits is not considered here because it is assumed that the carbon sequestration arising from the planted trees is instead used to offset on-farm carbon emissions that originate from livestock. Furthermore, in New Zealand, profit from forestry is only realised after approximately 25 years once the tree stand has matured and

is harvested. This time gap means the potential revenue from forestry does not support the farm cash flow at the time of the transition from pasture. For this reason, the potential long-term revenue from forestry operations was also not accounted for. Future development of the application should consider including the potential revenue from carbon credits and forestry.

## Landscape aesthetic and tacit knowledge

**Aesthetic.** The VR application combines high-resolution DEM (i.e., one meter), high-resolution aerial imagery (i.e., 300 mm) and realistic 3D tree models to provide highly realistic predictions of the landscape aesthetic (Fig 7). To improve the realism of the landscape texture, the aerial imagery has been augmented with a 'texture effect' that has been built within the UE and combines soil and grass effects (Fig 7F). The high level of image fidelity that was sought was expected to enhance the emotional emersion of the user and therefore a more holistic consideration of alternative future scenarios [4, 42, 43]. To preserve the performance of the application, the augmented texture is activated for the player's direct surroundings only. To preserve the performance of the application, the augmented texture is activated for the player's direct surroundings only. We believe VR applications similar to the one we have developed can simulate accurately the aesthetic value of different planting strategies and therefore contribute to landscape decision making [44].

Predicting the aesthetic dimension of potential future plantings is important when designing carbon neutral farms, because tree species have a significant effect on the aesthetic feel of a landscape and therefore, an individual's connection to it [45]. For instance, people tend to value forested landscape when seen from proximity, but in terms of landscape structure, preference for mosaic landscapes that have smaller areas of forest mixed with agricultural land has been reported [46, 47]. There is likely to be a large variation in an individual's landscape aesthetic preference which will reflect their own context, such as the place of residence and

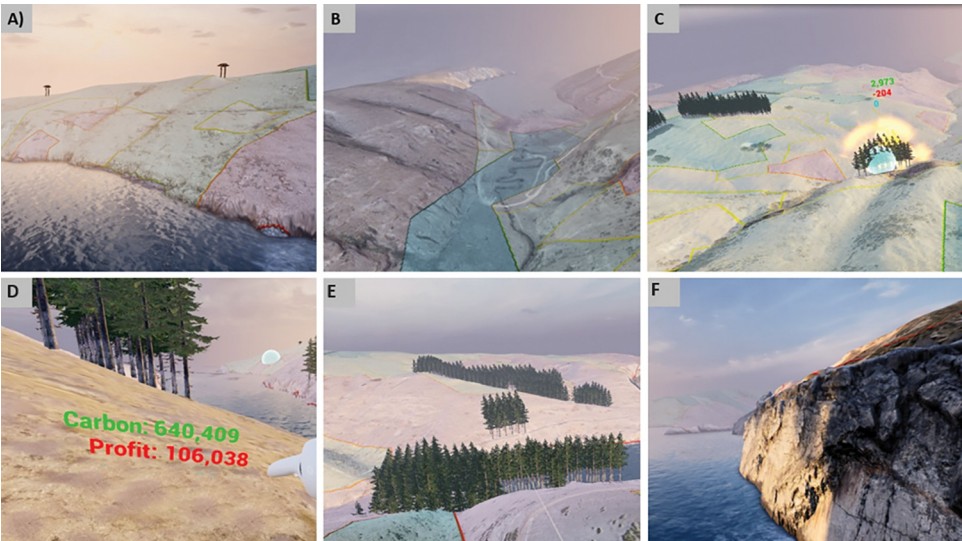

**Fig 7. Screen captures form the virtual reality application.** (A) and (B) helicopter views of two different areas of the farm. The coloured polygons on overlapping the pastures represent the different productivity classes. Dark green, light green, yellow and red polygons represent the very high, high, low and very low productive pasture respectively. (C) Represents trees being planted. (D) represents the predicted farm carbon emissions (in kg/year) and farm profit values (in $/year) and can be accessed by the user at any time. The aesthetic of Pinus radiata forest can also be assessed from the ground level. (E) Represents stands of Pinus radiata from a helicopter view. (F) represents the build texture that is used on top of the digital elevation model and the high-resolution aerial imagery to improve the realism of the rocky areas.

profession [47, 48]. In Washington (USA), researchers found that the planting sites that provided greatest benefits in terms of environmental outcomes were also the most suitable from an aesthetic point of view [49]. Similar observations have also been made in a study of urban-rural fringes in France [50]. However, there is limited empirical research on the role of trees in the aesthetic value of New Zealand rural landscapes to support the future development of landscape planning tools [51].

## Tacit knowledge

We have designed functionality that allows the user to virtually 'stand on the farm' as well as gain a wider perspective of the land from an aerial view. The ability to access both viewing perspectives is important for planning and for organising tacit knowledge (i.e., knowledge based on experience, practice and value) around the farm systems and management practices while interacting with the application.

For farm management tools that use mapping interfaces, an aerial view is often the default. These tools often rely on farm maps showing relevant features (e.g., paddocks) and associated attributes (e.g., area). An aerial view helps users to comprehend the spatial arrangement of land blocks, farm amenities and farm activities. With the uptake of new technology enabling precision agriculture, it is likely that the use of spatial data and maps within farm management tools will continue to progress [52]. At the same time, realistic visualisation specific to an individual land block helps to activate relevant tacit knowledge. It has been suggested that digital agriculture relying on robotics, sensors and big data will outperform existing farmers knowledge [53]. However, it has also been noted that our ability to collect data is currently exceeding our ability to turn it into actionable knowledge for farmers [53, 54]. In this context, it appears critical to find ways to combine the knowledge generated by decision support tools with the farmers tacit knowledge.

To stimulate the use of tacit knowledge we have developed a flexible motion system that allows the user to move efficiently within the virtual environment and to easily shift from an aerial view to a stand-on farm view. To move across the landscape, the application offers two locomotion options, including 1) a teleporting system where the user can use the hand controllers to point at a specific location and be teleported, and, 2) a '*free-movement*' system via the command analogue to move in any direction. This combination of locomotion systems helps to maximise the user's ability to explore large virtual landscapes (i.e., in this case 1,600 ha) efficiently while reducing the risk of motion sickness. Previous studies have demonstrated that users familiarity with an actual environment influences the way they perceive the virtual representation of the same environment. For instance, users who are familiar with a location have demonstrated a greater ability to assimilate changes that are made into the virtual world [55]. Further research is required to better measure the influence of tacit knowledge into the use of 3D geovisualisation tools. However, it is evident that digital twins create suitable virtual environments where users are able to combine their exisiting knowledge with new information displayed in a virtual sense.

The combination of remote sensing, land use data and VR has enabled a digital twin of a livestock farm in New Zealand to be created. We believe combining these types of technologies will be useful to help farmers understand trade offs arising from various planting strategies because farmers can reconcile their tacit knowledge with available scientific data. We believe this type of digital technology has considerable potential to enhance knowledge transfer to farmers, this includes integration of forestry and livestock farming to meet national emissions objectives.

VR applications, similar to the one we have developed, have the capability to depict the aesthetic outlook of different planting scenarios and therefore can support landscape decision

making, especially when consensus amongst people or compromise across tradeoffs is sort. There is also an ongoing trend to integrate existing virtual globe solutions such as Google Earth or Cesium with development platforms like UE [44]. Therefore, given its potentially valuable role and the advancements in the technology, we expect the development of digital twins (i.e., accurate 3D geovisulaisation) will become more common in the future.

## Conclusions

Planting trees on farms is an effective, and potentially profitable, way of lowering the carbon footprint of a farm enterprise. However, transitioning significant, or complete, portions of a farm into forestry has flow-on effects on farm profitability, landscape aesthetics and in many instances, rural communities as a whole. Therefore, on-farm planting strategies require careful selection of site and species to achieve the right balance across several competing factors.

We have developed a decision support mechanism using VR technology that draws together science data and user knowledge into a farm planning process that we believe will accelerate the transition of farming landscapes to a sustainable and low-carbon future. The VR application we have developed supports a realistic digital version of a farm that in turn helps elucidate the user's tacit knowledge into strategic planning without the need for sophisticated, and computer intensive models.

Under new carbon regulations in New Zealand, agricultural activities will strongly incentivise sheep and beef farmers to plant trees. While on first appearance this seems to be a relatively simple planning procedure, there remains confusion around the New Zealand ETS both at a farm and regional level. In our application we have enabled, either formally or informally, a diverse and complex combination of quantitative and qualitative factors (e.g., science data, spatial arrangement of amenities, cultural and aesthetic values, and personal connectivity) to be considered into the planning process. This project demonstrates that scientific data can be combined with VR to provide more holistic farm-scale assessments compared with the current suite of farm support tools.

In proving this concept, we have identified where additional work would improve the accuracy of the underpinning science data (e.g., GHG, financial and carbon sequestration), and increase the relevance of our approach to a wider set of considerations within the broader carbon-neutrality discussion. Future work might include a greater diversity of farms and farming situations, greater diversity of tree species, including a wider selection of native species, and an assessment of the user experience using VR applications compared with more traditional decision support tools. We have also identified a wider range of challenges to which our VR technology approach might be applied, such as on-farm biodiversity conservation and enhancement, mitigating contaminant (e.g., nutrient, microbe, and sediment) losses and new on-farm business development opportunities.

## Supporting information

**S1 Table. Carbon sequestration and establishment costs for the different tree species and carbon emissions and profitability for the different pasture production classes.**
(XLSX)

## Acknowledgments

The authors thank the owners of the land for enabling us to use their property as a pilot farm, and Ali Jansen for his contribution to the development of the virtual reality application.

## Author Contributions

**Conceptualization:** Remy Lasseur, Seth Laurenson, Mike Mackay.

**Data curation:** Remy Lasseur.

**Formal analysis:** Remy Lasseur.

**Funding acquisition:** Remy Lasseur, Seth Laurenson, Mike Mackay.

**Investigation:** Remy Lasseur, Mohsin Ali, Ian Loh.

**Methodology:** Remy Lasseur, Mohsin Ali, Ian Loh.

**Project administration:** Remy Lasseur.

**Software:** Mohsin Ali.

**Supervision:** Seth Laurenson.

**Visualization:** Mohsin Ali.

**Writing – original draft:** Remy Lasseur.

**Writing – review & editing:** Seth Laurenson, Mike Mackay.

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
