## [Decision Letter · Decision Letter 0]

3 Mar 2023

PONE-D-23-01762Designing profitable and climate-smart farms using virtual realityPLOS ONE

Dear Dr. Lasseur,

Thank you for submitting your manuscript to PLOS ONE. After careful consideration, we feel that it has merit but does not fully meet PLOS ONE’s publication criteria as it currently stands. Therefore, we invite you to submit a revised version of the manuscript that addresses the points raised during the review process.

We look forward to receiving your revised manuscript.

Kind regards,

Jun Yang

Academic Editor

PLOS ONE

Journal Requirements:

3. We note that Figure 7 in your submission contain copyrighted images. All PLOS content is published under the Creative Commons Attribution License (CC BY 4.0), which means that the manuscript, images, and Supporting Information files will be freely available online, and any third party is permitted to access, download, copy, distribute, and use these materials in any way, even commercially, with proper attribution. For more information, see our copyright guidelines: http://journals.plos.org/plosone/s/licenses-and-copyright.

a. You may seek permission from the original copyright holder of Figure 7 to publish the content specifically under the CC BY 4.0 license. 

4. We note that Figures 1, 4 and 6 in your submission contain map/satellite images which may be copyrighted. All PLOS content is published under the Creative Commons Attribution License (CC BY 4.0), which means that the manuscript, images, and Supporting Information files will be freely available online, and any third party is permitted to access, download, copy, distribute, and use these materials in any way, even commercially, with proper attribution. For these reasons, we cannot publish previously copyrighted maps or satellite images created using proprietary data, such as Google software (Google Maps, Street View, and Earth). For more information, see our copyright guidelines: http://journals.plos.org/plosone/s/licenses-and-copyright.

a. You may seek permission from the original copyright holder of Figures 1, 4 and 6 to publish the content specifically under the CC BY 4.0 license.  

Additional Editor Comments:

Reviewer 1

The authors designed profitable and climate-smart farms using virtual reality. The research methodologies are reasonable, and the findings are interesting. However, there are still some aspects that should be improved to make the paper publishable. I focus here only on some points, which are hopefully easy for the authors to take into account in the revision.

(1) Abstract - highlight the innovation and significance. In addition, 300 ha or 400ha, check it.

(2) Maybe data should be first introduced, and analysis Part Results.

(3) Legend is missing in some figures, add it.

(4) There are some references related to climate change and landscapes, I suggest you cited it in the manuscript, as follows.

1)Understanding seasonal contributions of urban morphology to thermal environment based on boosted regression tree approach, Building and Environment(2022), doi: 10.1016/j.buildenv.2022.109770.

2)The roles of surrounding 2D/3D landscapes in park cooling effect: Analysis from extreme hot and normal weather perspectives, Building and Environment(2023), doi: 10.1016/j.buildenv.2023.110053.

3)Impacts of urban green space on land surface temperature from urban block perspectives. Remote Sensing(2022) ,doi:10.3390/rs14184580.

4)Regional thermal environment changes: Integration of satellite data and land use/land cover, ISCIENCE (2023), doi: https://doi.org/10.1016/j.isci.2022.105820.

5)Relationship between urban spatial form and seasonal land surface temperature under different grid scales. Sustainable Cities and Society (2023), doi: https://doi.org/10.1016/j.scs.2022.104374

6)Contributions of sea–land breeze and local climate zones to daytime and nighttime heat island intensity. npj Urban Sustainability (2022) 10.1038/s42949-022-00055-z.

7)The impact of urban renewal on land surface temperature changes: A case study in the main city of Guangzhou, China. Remote Sensing (2020), doi: 10.3390/rs12050794.

8)Spatial expansion paths of urban heat islands in Chinese cities Analysis from a dynamic topological perspective for the improvement of climate resilience. Resources, Conservation & Recycling(2023),doi: 10.1016/j.resconrec.2022.106680.

9)The Research and Application of Virtual Reality (VR) Technology in Agriculture Science. Computer & Computing Technologies in Agriculture(2009)

10) Decision support tools for agriculture: Towards effective design and delivery. Agricultural Systems(2016), doi: 10.1016/j.agsy.2016.09.009.

Reviewer 2

The authors design a low carbon farms by virtual reality. The research is innovative, but it seems only like a design report. The problems need to be revised as follow. 1.The presentation is not standard enough. Firstly, in line 19-20, the tool combines virtual reality technology,environment science and high-resolution spatial data from an operational 300-hectare livestock farm. The environment science is a subject, it can not stand side by side with technology and data. The authors should illustrated environmental science theory, knowledge and so on. Secondly, the different tree means different types, height or others in line 62. Thirdly, the meaning of symbol should be illustrated behind Unreal Engine in line 69. 2.In line 73, there have other digital decision support tool. What's the difference of this VR with existing tools. What's the advantage and disadvantage? 3.The authors should use academic terms in this manuscript. To my knowledge is not belong to academic terms. The research should objective. 4.In 2.1 study site, the name of farm should be mentioned. Why the authors choose this farm? What's the characteristic or speciality of it? 5.In line 101-104, the authors introduced the structure of next section and section 3.3. It should be moved to the last paragraph in introduction. In addition, the structure of each section in this manuscript should be introduced detailed to improve the logic of different sections. 6.Some presentation is not clear enough. Such as line 130-133. 7.The fomulas are lack of serial numbers. 8.In line 139-147, this content is about figure 3 and have little relate to formula1, It should be moved to the end of 2.2. 9.The data source website of NDVI and Sentinel-2 should be supplied. 10.In line 172-174, the number of 4 categories ranging should be illustrated. 11.The structure of section 2 should be adjusted as follow. The logic is very mess. 2.Study site and Methods 2.1Study site 2.2 Data Source 2.3Methods 2.3.1 Carbon emissions and financial estimates 2.3.2 Virtual Reality application 2.4 Research framework 12.The authors should compare the image at present and design image. Then evaluate the profit improve and give some advices. 13.The 3 results and discussion should divide into two sections respectively. 14.The legend should supplied in figures. 15.Some presentation is a little blur in line 282-287. 16.In section 3.2.1 Aesthetic, the present is a little broad. It should illustrate how to VR design influence the aesthetic specificly. 17.In introduction, the authors should illustrate the background of climate change, land use and carbon emission. The references should be cited as follow. Response characteristics and influencing factors of carbon emissions and land surface temperature in Guangdong Province,China.Urban Climate,2022,46(51):101330.doi:10.1016/j.uclim.2022.101330. Regional thermal environment changes: Integration of satellite data and land use/land cover.iScience,2022,26,105820.doi:https://doi.org/10.1016/j.isci.2022.105820. Spatial and temporal heterogeneity of urban land area and PM2.5 concentration in China.Urban Climate,2022,45:101268.doi:https://doi.org/10.1016/j.uclim.2022.101268 Spatiotemporal relationship characteristic of climate comfort of urban human settlement environment and population density in China.Front. Ecol. Evol.2022,10:953725. doi: 10.3389/fevo.2022.953725 Spatio–te

Reviewers' comments:

Reviewer's Responses to Questions

**Comments to the Author**

1. Is the manuscript technically sound, and do the data support the conclusions?

Reviewer #1: Yes

Reviewer #2: Yes

2. Has the statistical analysis been performed appropriately and rigorously? 

Reviewer #1: Yes

Reviewer #2: Yes

3. Have the authors made all data underlying the findings in their manuscript fully available?

Reviewer #1: Yes

Reviewer #2: Yes

4. Is the manuscript presented in an intelligible fashion and written in standard English?

Reviewer #1: Yes

Reviewer #2: Yes

5. Review Comments to the Author

Reviewer #1: The authors designed profitable and climate-smart farms using virtual reality. The research methodologies are reasonable, and the findings are interesting. However, there are still some aspects that should be improved to make the paper publishable. I focus here only on some points, which are hopefully easy for the authors to take into account in the revision.

(1) Abstract - highlight the innovation and significance. In addition, 300 ha or 400ha, check it.

(2) Maybe data should be first introduced, and analysis Part Results.

(3) Legend is missing in some figures, add it.

(4) There are some references related to climate change and landscapes, I suggest you cited it in the manuscript, as follows.

1)Understanding seasonal contributions of urban morphology to thermal environment based on boosted regression tree approach, Building and Environment(2022), doi: 10.1016/j.buildenv.2022.109770.

2)The roles of surrounding 2D/3D landscapes in park cooling effect: Analysis from extreme hot and normal weather perspectives, Building and Environment(2023), doi: 10.1016/j.buildenv.2023.110053.

3)Impacts of urban green space on land surface temperature from urban block perspectives. Remote Sensing(2022) ,doi:10.3390/rs14184580.

4)Regional thermal environment changes: Integration of satellite data and land use/land cover, ISCIENCE (2023), doi: https://doi.org/10.1016/j.isci.2022.105820.

5)Relationship between urban spatial form and seasonal land surface temperature under different grid scales. Sustainable Cities and Society (2023), doi: https://doi.org/10.1016/j.scs.2022.104374

6)Contributions of sea–land breeze and local climate zones to daytime and nighttime heat island intensity. npj Urban Sustainability (2022) 10.1038/s42949-022-00055-z.

7)The impact of urban renewal on land surface temperature changes: A case study in the main city of Guangzhou, China. Remote Sensing (2020), doi: 10.3390/rs12050794.

8)Spatial expansion paths of urban heat islands in Chinese cities Analysis from a dynamic topological perspective for the improvement of climate resilience. Resources, Conservation & Recycling(2023),doi: 10.1016/j.resconrec.2022.106680.

9)The Research and Application of Virtual Reality (VR) Technology in Agriculture Science. Computer & Computing Technologies in Agriculture(2009)

10) Decision support tools for agriculture: Towards effective design and delivery. Agricultural Systems(2016), doi: 10.1016/j.agsy.2016.09.009.

Reviewer #2: The authors design a low carbon farms by virtual reality. The research is innovative, but it seems only like a design report. The problems need to be revised as follow.

1.The presentation is not standard enough. Firstly, in line 19-20, the tool combines virtual reality technology,environment science and high-resolution spatial data from an operational 300-hectare livestock farm. The environment science is a subject, it can not stand side by side with technology and data. The authors should illustrated environmental science theory, knowledge and so on. Secondly, the different tree means different types, height or others in line 62. Thirdly, the meaning of symbol should be illustrated behind Unreal Engine in line 69.

2.In line 73, there have other digital decision support tool. What's the difference of this VR with existing tools. What's the advantage and disadvantage?

3.The authors should use academic terms in this manuscript. To my knowledge is not belong to academic terms. The research should objective.

4.In 2.1 study site, the name of farm should be mentioned. Why the authors choose this farm? What's the characteristic or speciality of it?

5.In line 101-104, the authors introduced the structure of next section and section 3.3. It should be moved to the last paragraph in introduction. In addition, the structure of each section in this manuscript should be introduced detailed to improve the logic of different sections.

6.Some presentation is not clear enough. Such as line 130-133.

7.The fomulas are lack of serial numbers.

8.In line 139-147, this content is about figure 3 and have little relate to formula1, It should be moved to the end of 2.2.

9.The data source website of NDVI and Sentinel-2 should be supplied.

10.In line 172-174, the number of 4 categories ranging should be illustrated.

11.The structure of section 2 should be adjusted as follow. The logic is very mess.

2.Study site and Methods

2.1Study site

2.2 Data Source

2.3Methods

2.3.1 Carbon emissions and financial estimates

2.3.2 Virtual Reality application

2.4 Research framework

12.The authors should compare the image at present and design image. Then evaluate the profit improve and give some advices.

13.The 3 results and discussion should divide into two sections respectively.

14.The legend should supplied in figures.

15.Some presentation is a little blur in line 282-287.

16.In section 3.2.1 Aesthetic, the present is a little broad. It should illustrate how to VR design influence the aesthetic specificly.

17.In introduction, the authors should illustrate the background of climate change, land use and carbon emission. The references should be cited as follow.

Response characteristics and influencing factors of carbon emissions and land surface temperature in Guangdong Province,China.Urban Climate,2022,46(51):101330.doi:10.1016/j.uclim.2022.101330.

Regional thermal environment changes: Integration of satellite data and land use/land cover.iScience,2022,26,105820.doi:https://doi.org/10.1016/j.isci.2022.105820.

Spatial and temporal heterogeneity of urban land area and PM2.5 concentration in China.Urban Climate,2022,45:101268.doi:https://doi.org/10.1016/j.uclim.2022.101268

Spatiotemporal relationship characteristic of climate comfort of urban human settlement environment and population density in China.Front. Ecol. Evol.2022,10:953725. doi: 10.3389/fevo.2022.953725

Spatio–temporal evolution and factors of climate comfort for urban human settlements in the Guangdong–Hong Kong–Macau Greater Bay Area. Front. Environ. Sci.2022, 10:1001064. doi: 10.3389/fenvs.2022.1001064

6. PLOS authors have the option to publish the peer review history of their article (what does this mean?). If published, this will include your full peer review and any attached files.

Reviewer #1: No

Reviewer #2: No

---

## [Author Response · Author response to Decision Letter 0]

19 Apr 2023

15 April 2023 

Dear Professor Jun Yang

Thank you for considering our research paper, ‘Designing profitable and climate-smart farms using virtual reality. [PONE-D-23-01762]’.

I appreciate the time and effort the reviewers gave to reviewing our paper and I am grateful of the comments they have made and believe, with their comments addressed this paper has been strengthened. 

As the leading author, I have now worked through the reviewer comments and the manuscript carefully to addressed each of the points that was raised. 

Attached, please find the revised version of the paper. In the revision notes, I have listed actions taken in response to reviewer comments that were made to the original manuscript (Within this file, emendations are shown in red).

We, the authors, feel strongly that this paper is of value to the scientific community, and we hope the detailed changes to this paper will enable it to be published in PLOS ONE.

Yours sincerely

Remy Lasseur

AgResearch 

Lincoln Science Centre 

Private Bag 4749, Christchurch 8140 

New Zealand 

Phone: +64 21 904 357 .

E-mail: remy.lasseur@agresearch.co.nz

 

Editors Comments: 

File names have been edited to match the journal requirement. Figures have been run through PACE software to match PLOS ONE's standards. In the manuscript, figure labels as well as captions have been edited to match PLOS ONE's style requirements. The title page and the headings throughout the document has also been edited to match PLOS ONE's requirements.

All the data set underlying the results are in the public domain and the sources to access all the data sets are listed in the manuscript. This has been acknowledged in the feedback form the two reviewers (copied below). We have also added to the supporting information the look-up tables used for the carbon and financial modelling. Please note these data are similar in their entirety to what is currently included and presented in the manuscript. 

‘3. Have the authors made all data underlying the findings in their manuscript fully available?

The PLOS Data policy requires authors to make all data underlying the findings described in their manuscript fully available without restriction, with rare exception (please refer to the Data Availability Statement in the manuscript PDF file). The data should be provided as part of the manuscript or its supporting information or deposited to a public repository. For example, in addition to summary statistics, the data points behind means, medians and variance measures should be available. If there are restrictions on publicly sharing data—e.g., participant privacy or use of data from a third party—those must be specified.

Reviewer #1: Yes

Reviewer #2: Yes’

3. We note that Figure 7 in your submission contain copyrighted images. All PLOS content is published under the Creative Commons Attribution License (CC BY 4.0), which means that the manuscript, images, and Supporting Information files will be freely available online, and any third party is permitted to access, download, copy, distribute, and use these materials in any way, even commercially, with proper attribution. For more information, see our copyright guidelines: http://journals.plos.org/plosone/s/licenses-and-copyright.

Figure 7 show screen captures from the application we have developed. Therefore, they are owned by the authors and their institutions as a consequence of this study. On this basis we approve their release as a component of the publication being presented 

4. We note that Figures 1, 4 and 6 in your submission contain map/satellite images which may be copyrighted. All PLOS content is published under the Creative Commons Attribution License (CC BY 4.0), which means that the manuscript, images, and Supporting Information files will be freely available online, and any third party is permitted to access, download, copy, distribute, and use these materials in any way, even commercially, with proper attribution. For these reasons, we cannot publish previously copyrighted maps or satellite images created using proprietary data, such as Google software (Google Maps, Street View, and Earth). For more information, see our copyright guidelines: http://journals.plos.org/plosone/s/licenses-and-copyright.

All the layers used to generate the maps in figures 1, 4 and 6 are based on government data that is in the public domain and distributed under CC BY 4.0. The layers and their license can be found in the following url: 

• Digital Elevation Model: https://data.linz.govt.nz/layer/105027

• Land cover map: https://data.mfe.govt.nz/layer/52375-lucas-nz-land-use-map-1990-2008-2012-2016-v011/

• Aerial imagery: https://data.linz.govt.nz/layer/53519-canterbury-03m-rural-aerial-photos-2015-2016/

+++++++++++++++++++++++++++++

Reviewer: 1

1. Abstract - highlight the innovation and significance. In addition, 300 ha or 400ha, check it.

The abstract has been rewritten to highlight the significance of the research and its innovative aspect. As reviewer 1 as noticed there was an inconsistency in the manuscript regarding the area of the farm. The area of the farm is 400 hectare (and not 300 hectare). Lines 13 to 24.

2. Maybe data should be first introduced, and analysis Part Results.

We thank reviewer 1 for this comment and we have edited the structure of the second section of the manuscript to reflect the requested changes. Especially, we have added a sub-section to introduce the data before the analysis are described in the study site and methods section. Lines 116 to 158.

3. Legend is missing in some figures, add it.

Legends have been added in figures 1, 4 and 6.

4. There are some references related to climate change and landscapes, I suggest you cited it in the manuscript, as follows.

• 1) Understanding seasonal contributions of urban morphology to thermal environment based on boosted regression tree approach, Building and Environment(2022), doi: 10.1016/j.buildenv.2022.109770.

• 2) The roles of surrounding 2D/3D landscapes in park cooling effect: Analysis from extreme hot and normal weather perspectives, Building and Environment(2023), doi: 10.1016/j.buildenv.2023.110053.

• 3) Impacts of urban green space on land surface temperature from urban block perspectives. Remote Sensing(2022) ,doi:10.3390/rs14184580.

• 4) Regional thermal environment changes: Integration of satellite data and land use/land cover, ISCIENCE (2023), doi: https://doi.org/10.1016/j.isci.2022.105820.

• 5) Relationship between urban spatial form and seasonal land surface temperature under different grid scales. Sustainable Cities and Society (2023), doi: https://doi.org/10.1016/j.scs.2022.104374

• 6) Contributions of sea–land breeze and local climate zones to daytime and nighttime heat island intensity. npj Urban Sustainability (2022) 10.1038/s42949-022-00055-z.

• 7) The impact of urban renewal on land surface temperature changes: A case study in the main city of Guangzhou, China. Remote Sensing (2020), doi: 10.3390/rs12050794.

• 8) Spatial expansion paths of urban heat islands in Chinese cities Analysis from a dynamic topological perspective for the improvement of climate resilience. Resources, Conservation & Recycling(2023),doi: 10.1016/j.resconrec.2022.106680.

• 9) The Research and Application of Virtual Reality (VR) Technology in Agriculture Science. Computer & Computing Technologies in Agriculture(2009)

• 10) Decision support tools for agriculture: Towards effective design and delivery. Agricultural Systems(2016), doi: 10.1016/j.agsy.2016.09.009.

We thank reviewer 1 for the suggested literature. We have cited 2), 4), 5), 7), 9), 10) in the introduction and in the results and discussion section. Respectively reference number: 40, 36,39,38,17 and 3.

We have not cited 1), 3), 6), 8) as these papers present research that have a strong focus on urban areas and settlements that are very different to rural New Zealand.

+++++++++++++++++++++++

Reviewer: 2

1. The presentation is not standard enough. 

• Firstly, in line 19-20, the tool combines virtual reality technology, environment science and high-resolution spatial data from an operational 300-hectare livestock farm. The environment science is a subject, it can not stand side by side with technology and data. The authors should illustrated environmental science theory, knowledge and so on. 

We thank reviewer 2 for highlighting this inconsistency in the abstract. We have edited the sentence to remove the reference to 'environmental science'. Was read: 'The tool combines virtual reality technology, environmental science, and high-resolution spatial data from an operational 300-hectare livestock farm in New Zealand’. Lines 21 to 24, now reads: ‘For this proof-of-concept study, we incorporate virtual reality technology in Unreal Engine, environmental and financial data, and high-resolution spatial layers from an operational 400-hectare livestock farm in New Zealand.'

• Secondly, the different tree means different types, height or others in line 62. 

We have edited the sentence to provide the required information about the different tree species. Lines 63 to 65, now reads: ‘In practice, the VR application enables farmers to ‘stand in their farm’ (a virtual environment), and virtually plant different trees species in different locations. Different tree species have different carbon sequestration rates, aesthetic, and establishment costs. Therefore, the impacts of tree planting depend on the selected tree species and planting location.’

• Thirdly, the meaning of symbol should be illustrated behind Unreal Engine in line 69.

The initial manuscript had a registered trademark symbol next to Unreal Engine in line 69 presented as a superscript as follow: Unreal Engine®. It seems that the small size of the symbol might have caused confusion. To address this issue, we have used normal text as opposed to superscript for the symbol. Line 74, now reads: ‘Unreal Engine®’.

2. In line 73, there have other digital decision support tool. What's the difference of this VR with existing tools. What's the advantage and disadvantage?

We have provided additional details in the introduction to describe how the proposed application complement existing decision support tools.

Lines 76 to 87, now reads: ‘Several digital decision support tools are available in New Zealand that can be used to assist farmers to design carbon neutral farm enterprises [e.g., 12, 13]. We also acknowledge the presence of technical guidance documents that are freely available in New Zealand to help farmers understand the implications of converting land to forestry, such as look-up tables and technical guides. However, our new application complements the existing information by adding VR technology, a novel communication style seldomly used to convey scientific data [14]. We take this approach because the ability of humans to understand and assimilate information from pictures is greatly enhanced compared to conveyance through traditional scientific representations such as graphics and numbers [3, 13-17]. Therefore, we believe our approach will support more effective conversion of information into user knowledge and facilitate the implementation of new practices. So far, the use of visualisation in landscape planning has largely been limited to a supporting role for participatory processes or to help policy making [18, 19].’

3. The authors should use academic terms in this manuscript. To my knowledge is not belong to academic terms. The research should objective.

We assume the reviewer refers here to the following sentence in the introduction: ‘But, to our knowledge, it is the first attempt in New Zealand to utilise VR technology for farm design and, more specifically, strategic tree planting on agricultural properties’. 

None of the tool currently available in the market use virtual reality. So, we have edited the sentence. Lines 80 to 81, now reads ‘However, our new application complements the existing information by adding VR technology, a novel communication style seldomly used to convey scientific data [14].’

4. In 2.1 study site, the name of farm should be mentioned. Why the authors choose this farm? What's the characteristic or speciality of it?

We have edited the description of the study site to explain why this location has been chosen for this research. The added details complement the information that was presented in the initial manuscript.

Lines 99 to 103, now reads: ‘In choosing a pilot farm we looked for an operating livestock farm with a diverse terrain, that had the potential for tree planting. We also searched for a location that had high amenity value, in this case unabated views of the Pacific Ocean and access to recreational sites. At the time this study was initiated, the farm owners were embarking on a re-forestation programme that aimed to expand the area in native trees at the cost of the current pastoral land.’

However, we do not think that mentioning the name of the farm would add value or clarity for an international audience, therefore, we have not provided the name of the farm.

5. In line 101-104, the authors introduced the structure of next section and section 3.3. It should be moved to the last paragraph in introduction. In addition, the structure of each section in this manuscript should be introduced detailed to improve the logic of different sections.

Content initially in line 101 and 104 has been moved to the end of the introduction and details to introduce the different sections of the paper have also been added at the end of the introduction. Lines 90 to 95, now reads ' This paper focuses on the technical development of the application, beginning with a description of the methods we used to model farm profits and carbon emissions. The next section describes the study site and the technical modelling that we used to predict carbon emissions and profitability. The next section also describes the method we used to develop the VR environment and to visualise tree plantings and the impact of the planting regime on farm performance (carbon, profitability). We then present the outcomes delivered by the VR application and provide some reflections about how the tool might support more interactivity between users and scientific data used for farm design and to achieve carbon targets.'

6. Some presentation is not clear enough. Such as line 130-133.

The initial sentence highlighted by the reviewer was read: ‘Using the VR headset and hand controls, the user selects where the tree will be located (i.e., ‘planted’) within a digital twin of the farm developed from very high spatial resolution data. Trees can be planted only on areas specified as pasture and cannot be planted in the same location as buildings or other trees. A planting site is equivalent to a 30m x 30m block (also referred as cell hereafter), and the user can plant as many blocks as they like within the farm boundary.’

We assumed the lack of clarity raised by the reviewer here might have come from the invertible use of the words ‘cells’ and ‘blocks’. Therefore, to avoid the confusion, we have removed the word 'blocks' and only kept the words 'grid cell'. Thus, making clear that the matter of interest here are cells within a spatial grid.

Lines 180 to 182, now reads: ‘Within the VR twin of the farm and using the VR headset and hand controls, the user selects 30m x 30m grid cells where tree can be located. When the user validates a planting site, the whole farm profitability is estimated by equation (1).’

7. The fomulas are lack of serial numbers.

Serial numbers have been added to the equations.

8. In line 139-147, this content is about figure 3 and have little relate to formula1, It should be moved to the end of 2.2.

We thank reviewer 2 for this comment. The content has been moved to the discussion section (lines 305 to 314) which we believe enhance the clarity of the study site and method section.

9. The data source website of NDVI and Sentinel-2 should be supplied.

Sentinel-2 images have been accessed through the EOSDA LandViewer website. This information has been added in the manuscript. EOSDA and the LandViewer website has also been added to the paper’s references. Lines 121 to 122, now reads: ‘To do so, two satellite tiles from the 2019 lambing season (October and November) have been accessed through the EOS Data Analytics website [21].’

10. In line 172-174, the number of 4 categories ranging should be illustrated.

We thank reviewer 2 for these comments we have added a new figure to illustrate the value ranges (figure 2)

11. The structure of section 2 should be adjusted as follow. The logic is very mess.

2.Study site and Methods

2.1Study site

2.2 Data Source

2.3Methods

2.3.1 Carbon emissions and financial estimates

2.3.2 Virtual Reality application

2.4 Research framework

We thank reviewer 2 for the suggested structure. The structure of the method section has been changed to address the suggestion from reviewer 2. In particular, we added a section on data source with sub-section describing all the datasets used in the study. We have also moved the sub-section called ‘Virtual reality application’ before the sub-section ‘Carbon emissions and financial modelling’ which we believe makes the logic clearer.

Lines 116 to 158, now reads:

2. Study sites and Methods

2.1 Study site

2.2 Data source

2.2.1 Geospatial data

2.2.2 Carbon sequestration data

2.2.3 Carbon emission and profitability data

2.3 Methods

2.3.2 Virtual reality application

2.3.1 Carbon emissions and financial modelling

12. The authors should compare the image at present and design image. Then evaluate the profit improve and give some advices.

We understand that the reviewer might be calling for a case study where the optimal solution is demonstrated. However, given the near-infinite solutions that are possible we do not believe it is possible to ‘optimise’ in this sense. In this sense, optimal would reflect the user preference relative to the objectives being sought. Our tool enables the user to incorporate all their pre-determined preferences into the design – hence there is no one answer and indeed this is the reason why we see value in a tool of this type.

13. The 3 results and discussion should divide into two sections respectively.

Thank you very much for this recommendation. The original paper was actually written using the approach of results followed by discussion. However, we found that it led to significant reiteration of information because the study brings together several components that are not often combined i.e., geospatial data, virtual reality, science models. We believe the central theme is the product that represents the sum of these parts and the resulting outcomes of this sum –i.e., the tacit knowledge, aesthetic evaluation etc. Consequently, the results section would be extremely brief and lend itself to a reintroduction of these aspects in the discussion. Instead, we have re-arranged the results and discussion to include a more clearly defined breakdown of subheadings. 

We hope the editor can see the logic in our approach to keep the results and discussion combined.

14. The legend should supplied in figures.

Legends have been added in Fig 1, 4 and 6.

15. Some presentation is a little blur in line 282-287.

We have edited the description of the Fig 6 to make it clearer.

Was read: ‘Map of the farm border (red line) with the area that would need to be planted for the farm to be carbon neutral (grey pixels) when converting the least productive pixels first. Dark grey areas highlight the locations that would need to be planted if only pine forest is planted. Dark grey combined with light grey areas highlight the locations that would need to be planted if only native forest is planted’.

Lines 290 to 296, now reads: ‘Map of the farm border (red line). Grey shaded areas represent the footprint (area) that would need to be planted for the farm to be carbon neutral with the assumption that the area producing the least pasture are converted first. When considering pine trees, only the dark grey area would need to be planted for the farm to be carbon neutral. When considering natives trees, because their sequestration rate is smaller than pine trees, all the shaded area (dark and light grey) would need to be planted for the farm to be carbon neutral’.

16. In section 3.2.1 Aesthetic, the present is a little broad. It should illustrate how to VR design influence the aesthetic specifically.

We have addressed the reviewer’s comments by re-writing this section with greater reference to the role of VR in the designing carbon neutral farms. Lines 321 to 323 now reads: ‘The high level of image fedelity that was sought was expected to enhance the emotional emersion of the user and therefore a more holistical consideration of alternative future scenarios [4, 42, 43]’.

We have also added a paragraph at the end of the discussion lines 389 to 3905that reads ‘VR applications, similar to the one we have developed, have the capability to depict the aesthetic outlook of different planting scenario and therefore can support landscape decision making, especially when concensus amongst people or compromise across tradeoffs is sort. There is also an ongoing trend to integrate existing virtual globe solutions such as Google Earth or Cesium with development platforms like UE [44]. Therefore, given its potentially valuable role and the advancements in the technology, we expect the development of digital twins (i.e., accurate 3D geovisulaisation) will become more common in the future.’

17. In introduction, the authors should illustrate the background of climate change, land use and carbon emission. The references should be cited as follow.

• 1) Response characteristics and influencing factors of carbon emissions and land surface temperature in Guangdong Province,China.Urban Climate,2022,46(51):101330.doi:10.1016/j.uclim.2022.101330.

• 2) Regional thermal environment changes: Integration of satellite data and land use/land cover.iScience,2022,26,105820.doi:https://doi.org/10.1016/j.isci.2022.105820.

• 3) Spatial and temporal heterogeneity of urban land area and PM2.5 concentration in China.Urban Climate,2022,45:101268.doi:https://doi.org/10.1016/j.uclim.2022.101268

• 4) Spatiotemporal relationship characteristic of climate comfort of urban human settlement environment and population density in China.Front. Ecol. Evol.2022,10:953725. doi: 10.3389/fevo.2022.953725

• 5) Spatio–temporal evolution and factors of climate comfort for urban human settlements in the Guangdong–Hong Kong–Macau Greater Bay Area. Front. Environ. Sci.2022, 10:1001064. doi: 10.3389/fenvs.2022.1001064

We thank reviewer 2 for the suggested literature. We have cited 1), 2), 3) in the results and discussion section. Respectively reference number: 35,36 and 37.

We have not cited 4) and 5) as these papers present research that have a strong focus on urban areas and settlements that are very different to rural New Zealand.

---

## [Decision Letter · Decision Letter 1]

22 May 2023

Designing profitable and climate-smart farms using virtual reality

PONE-D-23-01762R1

Dear Dr. Lasseur,

We’re pleased to inform you that your manuscript has been judged scientifically suitable for publication and will be formally accepted for publication once it meets all outstanding technical requirements.

Kind regards,

Jun Yang

Academic Editor

PLOS ONE

Additional Editor Comments (optional):

Accept

Reviewers' comments:

Reviewer's Responses to Questions

**Comments to the Author**

1. If the authors have adequately addressed your comments raised in a previous round of review and you feel that this manuscript is now acceptable for publication, you may indicate that here to bypass the “Comments to the Author” section, enter your conflict of interest statement in the “Confidential to Editor” section, and submit your "Accept" recommendation.

Reviewer #1: (No Response)

Reviewer #2: All comments have been addressed

2. Is the manuscript technically sound, and do the data support the conclusions?

Reviewer #1: (No Response)

Reviewer #2: Yes

3. Has the statistical analysis been performed appropriately and rigorously? 

Reviewer #1: (No Response)

Reviewer #2: Yes

4. Have the authors made all data underlying the findings in their manuscript fully available?

Reviewer #1: (No Response)

Reviewer #2: Yes

5. Is the manuscript presented in an intelligible fashion and written in standard English?

Reviewer #1: (No Response)

Reviewer #2: Yes

6. Review Comments to the Author

Reviewer #1: (No Response)

Reviewer #2: The authors have revised the manuscript carefully. However, there still have some problems need to be revised as follow.

1.In introduction, the authors should illustrated the application of VR in existing researches.

2.In line 130, the carbon sequestration values for the different tree species should be introduced simply, the authors could list a table.

3.In line 196-198, the NDVI is represent vegetation coverage. While the authors use it to represent high productive land.It should be illustrated detailed.

4.How to define the steep slope, shallow soils and low fertility in this manuscript. The authors should build a index system and introduce the range of value.

5.The names of figures is too long. Some content could be shown in text. Such as figure 5,6 and 7.

6.Some sentence is repeat. Such as line 323-325. The authors should check the manuscript carefully.

7.The conclusion is too simple. The authors should illustrated how to plant tree can benefit to reduce the carbon emission and keep aesthetic scene in VR.

7. PLOS authors have the option to publish the peer review history of their article (what does this mean?). If published, this will include your full peer review and any attached files.

Reviewer #1: No

Reviewer #2: No

---

## [Editor Report · Acceptance letter]

26 May 2023

PONE-D-23-01762R1 

Designing profitable and climate-smart farms using virtual reality 

Dear Dr. Lasseur:

I'm pleased to inform you that your manuscript has been deemed suitable for publication in PLOS ONE. Congratulations! Your manuscript is now with our production department. 

Kind regards, 

on behalf of

Dr. Jun Yang 

Academic Editor

PLOS ONE